# Long-Term Care Resident Awareness and Interest in Spasticity Treatments

**DOI:** 10.3390/geriatrics6010021

**Published:** 2021-03-03

**Authors:** Mallory L. Hacker, Michael S. Putman, Chandler E. Gill, Maxim Turchan, Taylor S. Hudson, Amanda D. Currie, Fenna T. Phibbs, David Charles

**Affiliations:** 1Department of Neurology, Vanderbilt University Medical Center, Nashville, TN 37212, USA; turchanm@gmail.com (M.T.); fenna.phibbs@vumc.org (F.T.P.); david.charles@vumc.org (D.C.); 2Department of Physical Medicine and Rehabilitation, Vanderbilt University Medical Center, Nashville, TN 37212, USA; 3Department of Rheumatology, Northwestern University, Chicago, IL 60611, USA; msputman@gmail.com; 4Department of Neurological Sciences, Rush University Medical Center, Chicago, IL 60612, USA; chandler_e_gill@rush.edu; 5General Surgery, Duke University Hospital, Durham, NC 27710, USA; taylor.scott.hudson@gmail.com; 6Department of Medicine, Duke University Medical Center, Durham, NC 27710, USA; amandacurrie7187@gmail.com

**Keywords:** spasticity, neurologic disease, long-term care, patient preferences

## Abstract

Spasticity is common in long-term care settings (affecting up to one in three residents), yet it remains under-treated despite safe and effective, Food and Drug Administration (FDA)-approved therapies. One barrier to treatment may be lack of awareness of available therapies for long-term care residents living with spasticity. A standardized spasticity treatment awareness and interest interview was conducted with 18 nursing home residents and 11 veterans’ home residents in this cross-sectional study. Veterans’ home residents were also asked about potential barriers to receiving spasticity treatment. Many residents across both long-term care facilities were unaware of most of the treatment options for spasticity. Participants were most aware of physical/occupational therapy (83%, 95% CI: 65–93%) and least aware of intrathecal baclofen (21%, 95% CI: 9–39%). After learning about treatments, only 7% of participants (95% CI: 0–23%) were not interested in receiving any form of spasticity treatment. Among residents previously unaware of spasticity treatments, at least one quarter became interested in receiving treatment and at least one-fifth indicated possibly being interested in the treatment after learning about it. Potential barriers to receiving treatment included traveling to see a doctor and limited knowledge of insurance coverage of spasticity treatments. These results suggest that patient-centered approaches, including education and discerning patient preferences, may improve spasticity treatment in long-term care settings.

## 1. Introduction

Spasticity affects up to one-third of residents in long-term care settings but remains largely undiagnosed and undertreated [1,2,3,4]. This velocity-dependent increase in stretch reflex associated with muscle hypertonicity arises following disease or injury to the central nervous system (e.g., stroke). Symptoms of spasticity (e.g., abnormal limb positioning, muscle stiffness and weakness, and decreased dexterity) make performing activities of daily living (ADL) challenging, both for residents and caregivers, which may also lead to social embarrassment and stigmatization [3,5,6,7]. Untreated spasticity can result in additional problems, such as pain, contractures, limb deformity, and spasms, that also negatively impact quality of life [8,9].

Safe and efficacious treatments for spasticity include physical/occupational therapy, medications, and surgery. These treatments can enhance quality of life by improving care delivery and restoring ADL function [10,11]. Although there are a variety of treatments available for spasticity, it remains under-treated in long-term care settings [1,2,3,4,12]. One barrier to long-term care residents receiving spasticity treatment may be their lack of awareness of the range of available treatment options and their associated safety and efficacy profiles. This perspective regarding spasticity management has not yet been addressed, since there are no studies reporting long-term care resident awareness of or willingness to try available spasticity treatments. Therefore, to better understand patients’ perspectives regarding spasticity management, this study interviewed residents with spasticity in two long-term care facilities (a nursing home and a veterans’ home) to better understand their awareness of and preferences for spasticity treatments. Residents of long-term care facilities often have barriers to receiving specialty care, and veterans’ home residents were also asked an exploratory set of questions regarding potential barriers.

## 2. Materials and Methods

### 2.1. Setting

Participants in this study include residents of two long-term care facilities who participated in prior research projects and completed treatment awareness interviews as part of this investigation. At the first long-term care facility, the prevalence of spasticity in a single 240-bed university-affiliated nursing home was evaluated as part of a quality-improvement project conducted at the direction of the facility’s medical director (Vanderbilt IRB #071234) [3]. At the second long-term care facility, the prevalence of spasticity was also assessed in a 140-bed state-operated long-term care facility for veterans’ (Vanderbilt IRB #110470) [2]. All residents at both facilities determined to have spasticity by movement disorders specialists were approached to participate in the treatment awareness interview, and there were no exclusion criteria. In both long-term care facilities, written informed consent was obtained for all participants from either the resident or their designated medical decision maker (Vanderbilt IRB#071234, IRB#110470).

### 2.2. Treatment Awareness and Interest Interview

In both cohorts, participants with spasticity were asked to complete a treatment awareness interview, and interviews were conducted in-person at the long-term care facility by trained research assistants. The treatment awareness interview consisted of a standardized survey designed to evaluate the participant’s awareness of and willingness to consider each of five available spasticity treatments: oral medications, physical/occupational therapy (PT/OT), neurotoxin injections, intrathecal baclofen, and orthopedic procedures. Participants were asked if they were aware of each treatment (Yes, Unsure, or No). Operational definitions of each treatment were then read aloud. Specific names of therapies were excluded from definitions to focus the description on how the therapy would be administered and the most common problem associated with the treatment. Participants were then asked if they would be interested in possibly receiving each treatment (Yes, Maybe, Unsure, or No). 

Interviews conducted at the veterans’ home ended with an additional brief set of questions designed to evaluate potential barriers to treatment not captured in the treatment awareness interview. These questions covered five potential barriers: physician appointment scheduling, access to specialty care, transportation to physician offices, perceived efficacy of available therapies, and awareness of insurance coverage of available therapies.

### 2.3. Analysis

Survey data were analyzed using descriptive statistics. The Agresti-Coull method was used to determine 95% confidence intervals (CI) for proportions. Analysis was performed using STATA 15.1 (StataCorp. LP, College Station, TX, USA).

## 3. Results

### 3.1. Study Participants

Eighteen nursing home residents with spasticity (40%, 18/45) completed the treatment awareness interview. Most residents approached for an in-person interview provided informed consent and completed the interview (64%, 16/25), but among residents not competent to make medical decisions, only two medical decision makers provided informed consent (10%, 2/20). Nursing home participants were 61% female (11/18) with a mean age of 79.5 ± 14.1 years (Table 1). Eighty-nine percent of nursing home participants had upper limb spasticity (16/18), 56% had lower limb spasticity (10/18), and 44% had both upper and lower limb spasticity (8/18). As expected for this population, the majority of nursing home participants with potential spasticity etiology information available (9/18) had history of stroke (*n* = 7), while others had cerebral palsy (*n* = 1) and multiple sclerosis (*n* = 1). The majority of nursing home participants were not receiving any treatment for spasticity (89%, 16/18), while two were receiving oral baclofen.

Eleven veterans’ home residents with spasticity (79%, 11/14) completed the treatment awareness interview. Veterans’ home participants were all male (11/11) with a mean age of 77.5 ± 8.2 years (Table 1). None of the veterans’ home participants were receiving treatment for their spasticity.

### 3.2. Treatment Awareness and Interest Interview

Across both long-term care facilities (*n* = 29), participants were most aware of physical/occupational therapy (83%, 95% CI: 65–93%; Figure 1) and least aware of intrathecal baclofen (21%, 95% CI: 9–39%) and orthopedic procedures (34%, 95% CI: 20–53%). Three nursing home participants and one veterans’ home participant were not aware of any of the five spasticity treatments described. 

After hearing treatment definitions, only 7% of participants (95% CI: 0–23%) were not interested in receiving any form of spasticity treatment (10% (95% CI: 3–27%) were unsure). Participants across both facilities were most interested in physical/occupational therapy (66%, 95% CI: 47–80%) and least interested in therapies requiring surgery: intrathecal baclofen (24%, 95% CI: 12–42%) and orthopedic procedures (31%, 95% CI: 17–49%). Among residents previously unaware of spasticity treatments, at least one-quarter became interested in receiving treatment and at least one-fifth indicated possibly being interested in the treatment after learning about it (Figure 2). Among nursing home participants, women were more averse to surgical treatments than men (intrathecal baclofen: 55% (95% CI: 28–78%) vs. 28% (95% CI: 8–65%); orthopedic procedures: 72% (95% CI: 43–91%) vs. 28% (95% CI: 8–65%)).

### 3.3. Barriers to Treatment

Nearly two-thirds of veterans’ home participants thought that their quality of life would improve with spasticity treatment (64%, 95% CI: 35–85%; Table 2). More than half indicated that it was easy to get a doctor’s appointment if desired (55%, 95% CI: 28–79%), and most had the option to see a specialty doctor when necessary (82%, 95% CI: 51–96%). However, fewer than half thought it was easy to travel to the doctor’s office (45%, 95% CI: 21–72%), and fewer than one-third knew if their insurance covered any of the spasticity treatments (27%, 95% CI: 9–57%).

## 4. Discussion

This study is the first to report spasticity treatment awareness and interest among long-term care facility residents living with this common, treatable movement disorder. These results indicate that a large percent of residents with spasticity from two different long-term care settings (a nursing home and a veterans’ home) are unaware of most of the available spasticity treatment options but also that many are interested in the treatments once informed about them. Additional exploration of potential barriers to treatment revealed perceived challenges regarding travel to the doctor’s office and insurance coverage of spasticity treatments. 

Participants from both long-term care facilities had more awareness of than interest in receiving physical/occupational therapy, likely because these are commonly offered in long-term care settings and also in acute care settings for medical situations that often lead to long-term care placement. Participants’ interest in oral medications and neurotoxin injection exceeded their awareness of these treatments, and a majority of residents previously unaware of these treatments indicated interest or potential interest in receiving them after learning more about them. This result highlights a potential opportunity to expand education about these treatments to residents, their medical decision makers, and their healthcare providers. Across all spasticity therapies presented, more than one out of five previously unaware residents indicated they might be interested in receiving treatment, which suggests that further education may lead to receiving spasticity treatment. Both cohorts were more averse to spasticity therapies requiring surgery (intrathecal baclofen and orthopedic procedures) than other treatment options, consistent with patients preferring less invasive therapies for other conditions [13]. Female residents were more opposed to receiving invasive therapies than male residents, which is also consistent with gender preference reports for other neurologic disorders [14].

Spasticity is prevalent in long-term care facilities (up to one-third of residents) [1,2,3,4], and when present, it can interfere with the efficient provision of nursing care [3,5,6,7]. Spasticity can also be painful, which often limits activities of daily living and further increases the burden on the provision of care [8,9]. Despite these negative consequences of spasticity, there are several FDA-approved treatments available that can restore ADL function and care delivery [10,11]. The present study sheds light on one potential barrier to adoption of such therapies, with many residents of long-term care facilities living with spasticity being unaware of the treatments available to them. Importantly, after learning about available spasticity treatments, many residents were interested in receiving one or more of the available therapies. Therefore, these results highlight the opportunity to improve patient education concerning available spasticity treatments, which holds the potential to enhance care provided to many long-term care facility residents.

While this study reports similar results from residents of two long-term care facilities, the small sample size at each center limits the generalizability of these findings. Most veterans’ home residents with spasticity completed this interview (79%), compared to only 40% of nursing home residents with spasticity. This discrepancy in participation is attributed to the proportion of residents able to answer the survey at each facility, with nearly half of nursing home residents with spasticity requiring a medical decision maker in order to participate. The reliance on medical decision makers to conduct clinical research studies in long-term care settings remains a challenge [15]. Medical decision makers may also unintentionally represent a barrier to residents receiving spasticity treatments, due to their lack of awareness of available options. Care provider awareness of spasticity and its available treatment options was not evaluated, which may have also affected resident awareness and interest. Data regarding spasticity location and etiology were not collected in the veterans’ home cohort, and spasticity severity was not evaluated in either group. The depth of residents’ knowledge about treatment options was limited to yes/no responses, which could impact these findings. While this study focused on the patient perspective regarding awareness and interest in several spasticity treatments, not every spasticity treatment will be indicated for every long-term care resident.

## 5. Conclusions

These results suggest that many long-term care residents living with spasticity are unaware of all treatment options available to them but are interested in receiving spasticity treatments after learning about them. More work is needed in a larger population to evaluate long-term care resident awareness of and interest in the wide variety of safe and effective spasticity treatments available. A more patient-centered approach that combines patient, medical decision maker, and provider education with informed patient preferences will likely improve access to spasticity treatment in long-term care settings.

## Figures and Tables

**Figure 1 geriatrics-06-00021-f001:**
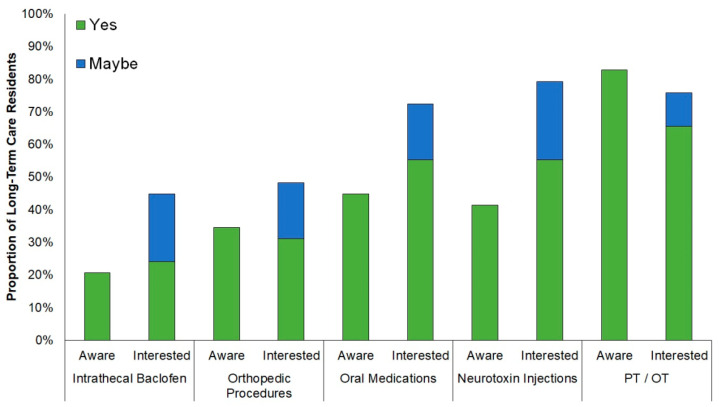
Long-term care facility resident awareness of and preferences for spasticity treatments. Twenty-nine residents with spasticity in two long-term care facilities (nursing home, *n* = 18; veterans’ home, *n* = 11) were first asked about their awareness of each spasticity treatment. Operational definitions were then read aloud, and residents were asked if they would be interested in getting each treatment. Responses not shown to questions regarding interest—“No”: intrathecal baclofen (45%), orthopedic procedures (52%), oral medications (21%), neurotoxin injection (17%), PT/OT (21%); “Unsure”: intrathecal baclofen (10%), oral medications (7%), neurotoxin injections (3%), and PT/OT (3%). PT/OT = Physical Therapy/Occupational Therapy.

**Figure 2 geriatrics-06-00021-f002:**
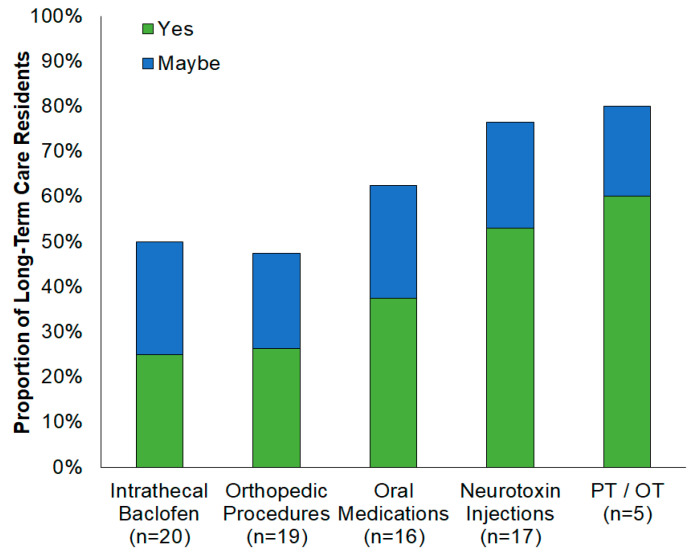
Impact of patient education on interest in spasticity treatments. Proportions of residents previously unaware of spasticity treatments (*n*) who subsequently indicated interest (Yes) or potential interest (Maybe) in receiving treatments after learning about them. Responses not shown to questions regarding interest—“No”: intrathecal baclofen (50%); orthopedic procedures (53%); oral medications (25%); neurotoxin injections (18%); PT/OT (20%); “Unsure”: oral medications (13%); and neurotoxin injections (6%).

**Table 1 geriatrics-06-00021-t001:** Participant characteristics.

	Nursing Home	Veterans’ Home
Participants, *n*	18	11
Age in Years Mean ± SD (Age Range)	79.5 ± 14.1 (51.4 to 100.3)	77.5 ± 8.2 (66.5 to 91.1)
Sex, *n* (%)		
Male	7/18 (39%)	11/11 (100%)
Female	11/18 (61%)	0/11 (0%)
Spasticity Being Treated, *n* (%)	2/18 (11%)	0/11 (0%)

**Table 2 geriatrics-06-00021-t002:** Potential barriers to spasticity treatment for long-term care residents.

**Question, % (*n*)**	**Easy**	**Sometimes Difficult**	**Difficult**	**Unsure**
How easy is it for you to get a doctor’s appointment when you want one?	55% (6)	9% (1)	18% (2)	18% (2)
How easy is it for you to get from where you live to doctor’s office?	45% (5)	18% (2)	9% (1)	27% (3)
**Question, % (*n*)**	**Yes**		**No**	**Unsure**
Do you have the option to see a special doctor, like a heart doctor or brain doctor, if you want?	82% (9)		18% (2)	0% (0)
Do you think getting treated for spasticity would make your quality of life better?	64% (7)		36% (4)	0% (0)
Do you know if your insurance covers any of the treatments?	27% (3)		9% (1)	64% (7)

Veterans’ home participants (*n* = 11).

## Data Availability

The data presented in this study are available from the corresponding author upon reasonable request.

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
