# Peer review of "Long-Term Care Resident Awareness and Interest in Spasticity Treatments"

_geriatrics, 2021, doi:10.3390/geriatrics6010021_

Round 1
Reviewer 1 Report
General comments
The authors conducted a study on the awareness of and the interest in spasticity treatments among long-term care residents. In general, the manuscript is very well written. The study aims are clearly defined, methods and results are clearly presented and conclusions are justified.
There are some major limitations that need to be considered. The sample size is small and the representativeness of the sample is questionable. The design neglects that not every treatment might be indicated in every participant (or if contraindications to certain treatments might be present); i.e., a movement disorders specialist has assessed whether spasticity is present (or not), but it wasn’t assessed which treatments were indicated in the respective patient (at least this is not reported). There is also no additional medical information given on spasticity, e.g. on location or etiology. These limitations should be addressed in detail, including their consequences for interpretation.
Specific comments/suggestions
Results section:
- page 2, line 86: if available, please give more detailed information on the type of spasticity.
- page 2, line 92: please give the kind of treatment of those treated for spasticity
- page 3 and 4, sections 3.2 and 3.3: It should be considered to give 95% confidence intervals to the presented percentages. This would illustrate the uncertainty resulting from the small sample size.
Reviewer 2 Report
Thank you for your article. I am recommending some very minor changes, which I hope you find helpful.

Round 2
Reviewer 1 Report
Thank you to the authors for performing the suggested changes to the manuscript. I have no further comments.